# Taxonomy of Defects in Auxiliary Elements of Facades and Its Relation with Lawsuits Filed by Property Owners

**Manuel J. Carretero-Ayuso** [1], **Carlos E. Rodríguez-Jiménez** [2,*] , **Maria Teresa Pinheiro-Alves** [3] **and Enrique Fernández-Tapia** [4]

1   Musaat Foundation and Department of Architecture, University of Alcalá, 28801 Alcalá de Henares, Spain; carreteroayuso@yahoo.es
2   Department of Architectural Construction II, University of Seville, 41012 Seville, Spain
3   Department of Architecture, University of Évora, 7000-811 Évora, Portugal; tpa@uevora.pt
4   Department of Architecture, University of Alcalá, 28801 Alcalá de Henares, Spain; fernandez.tapia@uah.es
*   Correspondence: ceugenio@us.es

**Abstract:** While at first it can be thought that the auxiliary elements of facades are merely ornamental with little practical function, this study shows that these components of the building envelope have a high impact on the envelope's functioning and performance. This is carried out through the analysis of all relevant lawsuits filed in Spain over a 10 year period, a data set in which a surprisingly high number of 1033 cases of defects was found to affect external windowsills, exterior wainscots or cornices (the three auxiliary elements considered) was found. Considering the total number of lawsuits, this is an objectively unprecedented study. An analysis is carried out regarding the interrelations between elements, defects, causes, and types of buildings, with the aim of obtaining a sorted classification of the data. This constitutes a useful tool to prevent future problems arising from either the design, execution, or maintenance of facades. These include various issues resulting from humidities, one of the most frequent envelope defects, which are found to significantly affect the auxiliary elements of facades.

**Keywords:** forensic engineering; exterior wainscots; cornices; external windowsills; damages





## 1. Introduction

The present study on lawsuits related to auxiliary elements of facades focuses on three large groups of construction units: exterior wainscots (*horizontal strips protecting the lower part of facades*), cornices (*linear highlights on the facade that have a decorative and finishing purpose*) and external windowsills (*water drainage device placed with a slope in the lower part of facade openings*). This is a set of integrated components in building facades, and they have important active/protecting functions, in addition to decorative ones. Existing research into these elements falls under a novel line of study that could be referred to as 'research niche' since specific references are extraordinarily limited are do not usually constitute the main object of the scientific analysis on construction elements. When a reference is made to them, it is usually tangential and brief within the context of wider studies on other facade elements such as window openings [1] and their perimeters [2,3]. At times they are referred to in studies on roofs, specifically with regard to their tail-end with the facades and drainage systems they have in common [4], as well as in studies on anomalies of envelopes [5] or of buildings as a whole [6].

While at first glance they might be considered to be secondary, the present study makes clear that, in most cases, these auxiliary elements perform important functions in the facade as a whole, impacting both the habitability and sustainability of the building. In this regard, their role as protective elements can be highlighted [7]—it being necessary to consider them in any analysis of the causes of defects occurring in facades [8]. These include construction problems related to water drainage and humidities in the facade, an aspect in which many

of these auxiliary elements have a clear and significant mission as a barrier against rain infiltrations [9], as they contribute to rainwater drainage and prevent water from affecting specific zones, minimising the appearance of humidities [10]. They thus contribute to the optimisation of the facade's hygrothermal performance, which in turn improves the thermal transmittance [11]. Additionally, as these elements are often constructed with materials that are similar to the rest of the facade, they also have a significant role in mechanical problems that may appear, especially in the case of fissures [12]. Lastly, they have an undeniable aesthetic function as they are usually visible on the facade.

These complementary and auxiliary elements, then, can be appropriately regarded as essential components of facades, and as such, their repercussion in the main indicators of building sustainability (functionality, aesthetics, comfort, and safety) is clear [13]. As such, the study of these facade construction units should be carried out with the same rigour and premises of any other element, and the same criteria are applied.

Among the aspects that are important in the facade as a whole, durability stands out since facades' direct exposure to the external environment implies that the evaluation of their service life is always important [14]. In this regard, an example study on the service life of windows was carried out in Portugal (a country whose construction traditions are similar to those in Spain) [15]. It found that the external factors most likely to affect the prediction of facades' service life are orientation, exposure to water/wind, distance to the sea, and pollution. To these environmental factors should also be added maintenance conditions. Other studies in warm climates also include sun exposure as having a direct relation with the most common defects found in facades [16].

Furthermore, in light of climate change, it has become necessary to place added focus on environmental factors since these will have a greater impact and will increase the loads resulting from meteorological phenomena (such as rain or wind). The good sizing and execution of auxiliary elements can be crucial so that these may perform as required and to avoid the subsequent appearance of defects and problems, such as is demonstrated when studying envelope drainage [17] or in the analysis of box gutters to prevent overflow as a result of rainwater [18]. The studies on the effects of wind [19] or of wind-driven rain [20], usually focused on the structure, also demonstrate the effect of these phenomena on facade components.

Maintenance is, without a doubt, another key factor for buildings and their envelopes, and it should always be considered as a means of reducing the probability of failures and improving performance [21], including simpler maintenance operations such as cleaning [22]. Along with maintenance, repair activities are necessary to maintain the performance of buildings and are always relevant during their period of use, particularly affecting facades and their components. For this reason, it is helpful if regular diagnosis and action plans are made available for the auxiliary elements of facades so as to carry out suitable repairs whenever any interventions are planned for the building envelope [23]. In addition, and given the uniqueness of many of these interventions, their corresponding proposals should include specific indicators that will allow an evaluation of the improvements in question and optimise investments [24] (as is the case with parameters of $CO_2$ emission reductions or the reduction in the demand for energy rehabilitations). In any case, when addressing maintenance, repair, or rehabilitation works, it is important to keep in mind that these involve economic repercussions for the building owners throughout the building's service life, for which reason investments are often reduced due to circumstantial budgetary reasons. In order to overcome this obstacle, it must be kept in mind that it has been demonstrated that, in the long run, these operations represent cost savings by preventing failures and mitigating the need for subsequent large-scale interventions [25].

It is also worth mentioning historical buildings, where interventions in facade elements require special attention, since they mostly refer to special circumstances pertaining to deterioration, construction systems, or patrimonial value. In this way, the common problems of humidities and salinity are often more complex [26] and should be addressed with specialised techniques, such as in facades with stone [27,28]. As such, these interventions

should include suitable planning from the start of the process, with design procedures marked by attention to detail [29]. In any case, regarding auxiliary elements, the design phase must always include high precision and detail, both in rehabilitations and in new buildings, since insufficient attention to any of these units in this phase will always increase the probability of subsequent appearance of defects [30]. It is thus concluded that continuous reviews and improvements to design procedures are highly recommended retro-feedback measures [31].

In conclusion to this chapter, it is worth highlighting the uniqueness of this research as to its data source, which contains the totality of lawsuits pertaining to auxiliary elements in the period and country considered (Spain). This corresponds to a high number of records that cover the entire scope of the matters under study, overcoming the primary limitation of similar studies: access to data (which is the fundamental problem for research) [21]. Moreover, the fact that the data proceed from lawsuits ensures the independence and absence of any patterns in the data collection. The objective of this study, then, is to know and analyse the defects that take place in the auxiliary elements of facades, determine their frequency, and quantify the underlying causes.

## 2. Materials and Methods

### 2.1. Researched Data Collection

The data source used in this study contains the records of an insurance company: the civil responsibility insurance company of building engineers in Spain [32]. All these records refer to lawsuits filed by building owners between 2008 and 2017 [33] as a result of defects appearing in auxiliary elements of facades (exterior wainscots, cornices, and external windowsills).

Below are shown photographs of some cases of defects found by the authors during the execution of works (see Figure 1), included as a visual addition to this research.

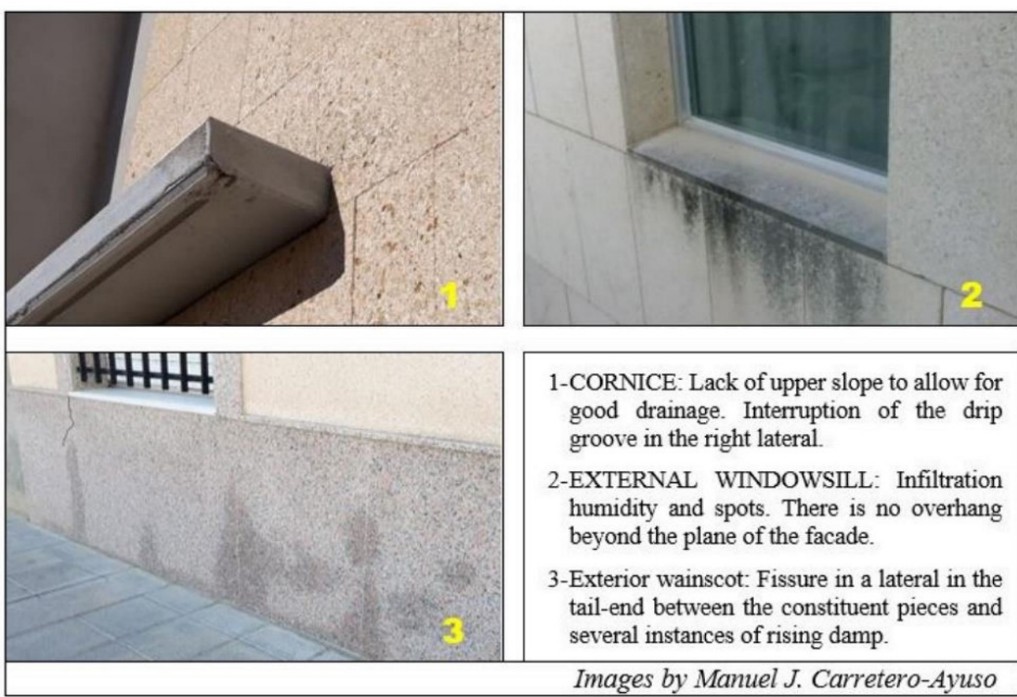

1-CORNICE: Lack of upper slope to allow for good drainage. Interruption of the drip groove in the right lateral.

2-EXTERNAL WINDOWSILL: Infiltration humidity and spots. There is no overhang beyond the plane of the facade.

3-Exterior wainscot: Fissure in a lateral in the tail-end between the constituent pieces and several instances of rising damp.

*Images by Manuel J. Carretero-Ayuso*

**Figure 1.** Photographic examples of defects in the three types of elements.

### 2.2. Characteristics of the Data

The review and verification of these judicial records yielded 1033 cases related to auxiliary elements of facades. They are the result of the collection, grouping, and typification of the entire population of lawsuits filed by owners and assessed by the forensic experts

that produced the technical reports presented to the Spanish Administration of Justice. It is not, then, a mere checklist. The initial detection is carried out by owners themselves, and, when their services are engaged, experts verify these defects to confirm their existence and magnitude. Once the lawsuits in question are considered final (i.e., unappealable)—several years later—the corresponding data are taken into account for study and evaluation in this paper.

The method used in this research is neither a 'case study', an 'experiment', nor a 'survey'. It is an analysis of the entirety of the population of these cases in the whole of Spain.

For the purposes of this study, the methodology of the Spanish regulation [34] on the assessment, identification, and evaluation of types of defects and types of causes was followed.

The collection of data used in this research corresponds to 100% of the existing cases, both with regard to the number of cases and to the number of years under analysis (there not being any uncertainty, given that it is not a partial sample). These are represented in Figure 2.

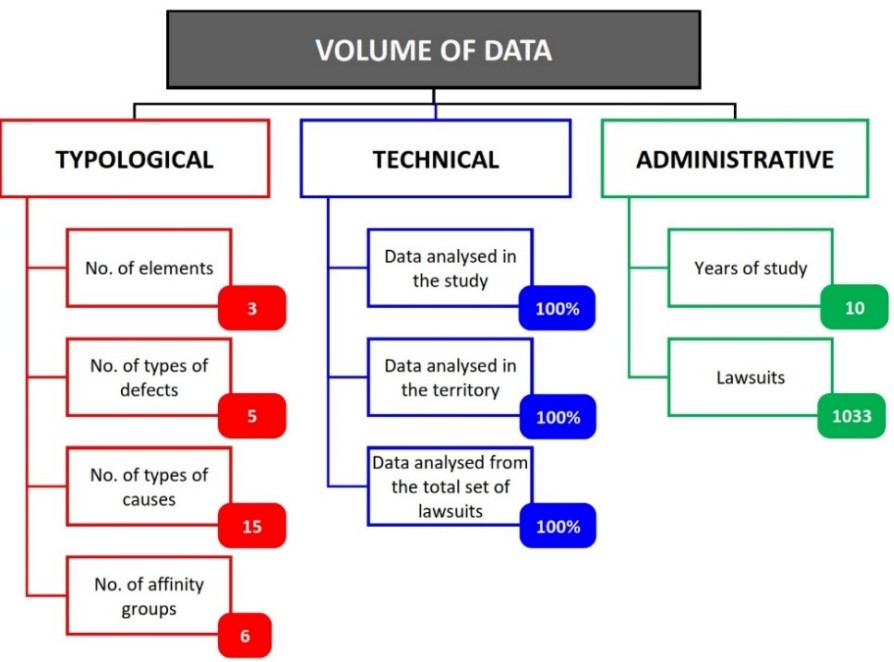

**Figure 2.** Amount of data used in the research.

### 2.3. The Spanish System of Law and Responsibilities

In Spain, technicians (architects and engineers) who produce designs and run and control the process of building construction are legally and functionally independent from the developer (the company one that makes the investment and promotes the project) and from the building company (which carries out the works in and of themselves). These technicians are required by the Spanish state to have a university degree, while the two aforementioned companies do not have a legal obligation of employing technicians (though they often do so for larger construction projects). This latter fact, combined with an organisational system that mostly subcontracts different trades, frequently leads to works being conducted with insufficient professional knowledge or a lack of proper responsible involvement in the construction process. As a result, there is a greater likelihood of problems or anomalies emerging on-site, with frequent issues of lack of quality and incorrect application of regulations.

There is another important aspect that distinguishes technicians from those companies. Technicians (architects and engineers) running the works are required by law to have civil responsibility insurance, while the developer and construction companies are not. As

such, when owners detect a defect and reach out to one of these companies (especially the construction company) to resolve it, the company will often decline to assist—especially if the repairs are costly—arguing that legal responsibility for the defect is debatable. Owners are then forced to resort to courts of law to defend their rights.

To explain the characteristics of the damages to the judge, owners hire a forensic expert to produce a report detailing the characteristics and scope of the damages. Once it is established that those problems do indeed exist in the houses/buildings in question, the judge may rule that the defects must be repaired and or the owners should be indemnified. Responsibility is often shared in such cases, but the fact that technicians have mandatory civil responsibility insurance often leads to the repair costs being defrayed by the insurance companies (of the technicians). Indeed, to achieve this outcome, the developer and construction companies will sometimes declare bankruptcy as a means to avoid being responsible for the thousands of euros needed to repair the defects.

*2.4. Study Procedure*

The bibliographic references reviewed have a heterogeneous and diverse procedural basis, as pointed out in the introduction of this paper. In this case, the inductive procedure is adapted to the specific conditions of the data source used—lawsuits filed in Spain. It should be noted that no methodological precedents were found in other countries in which a comparable data source was used for the study of auxiliary elements of facades. The difficulty in accessing and obtaining permission to know the contents of this type of official documents, as well as in obtaining a sufficiently high number of cases, is probably the reason for which no precedent was found for a comparable study. In other words, this research constitutes a novel contribution to the field of forensic engineering.

In the international literature that was reviewed [30,35,36], it was observed that a significant proportion of existing research on construction defects focuses on buildings that were financed by a specific developer or built by a specific construction company. In addition, studies often consider only a single building or a small set of buildings, for which reason their results have some amount of bias. In the present study, there is no parameter that might—for one reason or another—tie the cases between them, as the data consist of judicial records without any dependencies between them.

Below are defined concepts that intervene in the process of deterioration or appearance of defects, referred to as the 'indicators' of the study. Four such concepts were considered: the 'types of elements' or construction units (E), the 'types of defects' (D), the 'types of causes' (C) that lead to them, and the 'types of buildings' (B) wherein they appear.

Table 1 shows the classification system established for these indicators, grouping them in 'affinity groups'. The specific concepts of each group were also determined, and they were assigned identification codes. In this way, 5 different 'types of defects' were obtained, each one belonging to an 'affinity group of defects' (D/AG) that connects them according to similarities in their characteristics. In turn, 15 'types of causes' were obtained, each being assigned to an 'affinity group of causes' (C/AG). Lastly, 3 'types of building' were considered, without any affinity group.

The idea behind the above categorisation is to obtain a quadrant of construction interrelations between the defects that normally occur in these elements and the possible causes behind them. Some examples of these types of interrelations have been used in research on brick walls [37] and roofs [38]. The creation of these types of quadrants of construction interrelations is not in any way intended to replace the professional experience of technicians nor their judgment/assessment of each case. The idea is to provide basic information that can then be accessed by less experienced individuals once it has been validated through this work (volume and types of data shown Figure 2).

**Table 1.** Classification of the indicators, their affinity groups, and the underlying concepts.

| Indicator | Affinity Group | Code | Concept |
|---|---|---|---|
| Types of Elements (E) | — | E1 | Cornices |
| | | E2 | External windowsills |
| | | E3 | Exterior wainscots |
| Types of Defects (D) | Cleaning (D/AG-C) | D1 | Existence of spots and dirt |
| | Breaks (D/AG-F) | D2 | Fissures originating in the construction process |
| | | D3 | Detachment and fracture of pieces |
| | Humidities (D/AG-H) | D4 | Rising damp |
| | | D5 | Infiltration humidity |
| Types of Causes (C) | Incorrect conditions of execution (C/AG-I) | C01 | Lack/anomaly of drip groove, gutter, and/or roof downpipes |
| | | C02 | Lack/insufficiency of sealant |
| | | C03 | Lack/insufficiency of slope |
| | | C04 | Lack/anomaly in the waterproofing |
| | | C05 | Lack/insufficiency of moistening of the base |
| | Conditions of contact between elements (C/AG-E) | C06 | Lack of barrier against capillary humidity |
| | | C07 | Direct contact with the terrain |
| | | C08 | Lack of construction joints or expansion joints |
| | | C09 | Deterioration by humidity |
| | | C10 | Presence of water table |
| | | C11 | Lack or separation between pieces |
| | Conditions related to the materials (C/AG-M) | C12 | Incorrect anchoring material |
| | | C13 | Characteristics of the material used |
| | | C14 | Lack of adherence to the base |
| | | C15 | Poor quality of the previous render |
| Types of Buildings (B) | — | B1 | Flats |
| | | B2 | Houses |
| | | B3 | Other buildings |

Table 2 shows a quadrant of construction interrelations between types of defects and their possible causes, using a system of interrelation and classification that intends to standardise the categorisation and naming of the concepts involved. This approach is based on reading different studies related to reports on anomalies [39,40] and to materials [41–46], as well as on the authors' prior experience in this type of intervention.

The five types of defects (D) are shown in the columns of the table, and the fifteen types of causes (C) are shown in its rows. Cells indicate the degree of interrelation between them, or D–C.

Several studies [47–50] indicated the existence of specific defects as being predominant in comparison to other, less frequent, defects. The present study thus aimed to identify which defects occur most often in auxiliary elements of facades and which, in turn, occur less frequently.

In order to consider the interrelations proposed in Table 2, it was thought of obtaining and accessing a data source with two essential characteristics: sufficient volume and sufficient prestige (i.e., that it be irrefutable). In other words, it should have a large volume of detailed and reliable information. It would then be necessary to establish the existence (or lack) of each construction interrelation, as shown in the interrelations quadrant. As

stated at the beginning of the methodology section, this data source was the set of lawsuits filed by owners.

**Table 2.** Quadrant of construction interrelations between the types of defects and the types of causes.

| | | TYPES OF DEFECTS | | | | |
| --- | --- | --- | --- | --- | --- | --- |
| | | D/AG-C | D/AG-F | | D/AG-H | |
| | | D1 | D2 | D3 | D4 | D5 |
| **TYPES OF CAUSES** | **C/AG-I** | | | | | |
| | | C01 ● | – | – | – | ○ |
| | | C02 – | – | – | – | p●/f○ |
| | | C03 – | – | – | – | ○ |
| | | C04 – | – | – | – | ○ |
| | | C05 – | – | ○ | – | – |
| | **C/AG-E** | | | | | |
| | | C06 – | – | – | ● | – |
| | | C07 – | – | – | ○ | ○ |
| | | C08 – | p○/f● | ○ | – | – |
| | | C09 ○ | – | – | ○ | ○ |
| | | C10 – | – | – | ○ | – |
| | | C11 – | – | ○ | – | – |
| | **C/AG-M** | | | | | |
| | | C12 – | ○ | ● | – | ○ |
| | | C13 p○/f● | ● | ○ | – | – |
| | | C14 – | ○ | – | – | – |
| | | C15 – | ○ | ○ | – | – |

Legend: – = There is no construction interrelation between the type of defect and the type of cause. ○ = There is an interrelation between the type of defect and the type of cause, and this is not very recurrent (fewer than 1 out of each 3 times). ● = There is an interrelation between the type of defect and the type of cause, and this is quite recurrent (more than 1 out of each 3 times). p○/f● = Interrelation that was previously considered to be '○' and, following the validation of the study, was found to be '●'. p●/f○ = Interrelation that was previously considered to be '●' and, following the validation of the study, was found to be '○'.

In addition to all of the above, the authors took note of the geographical location of the buildings to correlate the data with a series of data unrelated to the judicial records. In this way, for each case, the type of climate (Oceanic, Continental, or Mediterranean) in that location was identified, as was the relative latitude in the country of Spain (North, Central, or South).

## 3. Results

Below are shown the results of the different analyses carried out on the 1033 cases in scope.

### 3.1. Results by Type of Element

Figure 3 shows the percentages of defects occurring in each type of element. It can be seen that over half the cases occur in external windowsills (E2 = 51.98%), followed by exterior wainscots, which aggregate over a third of all cases.

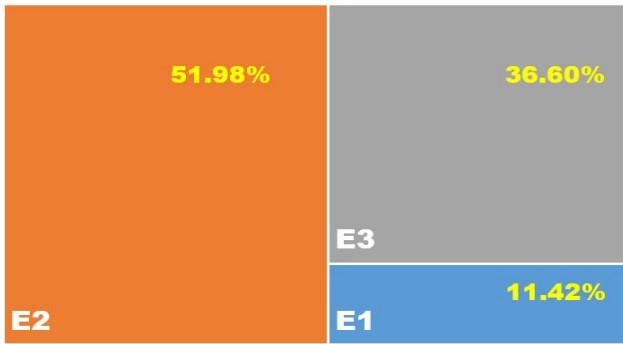

**Figure 3.** Percentage of presence by type of element.

*3.2. Results by Type of Defect*

Figure 4 shows the recurrence of each defect. The most frequent defect is 'infiltration humidity' (D5 = 50.72%), followed by 'detachment and fracture of pieces' (D3 = 21.30%), with 'rising damp' in third place (D4 = 16.94%). These three defects account for around 90% of the total cases.

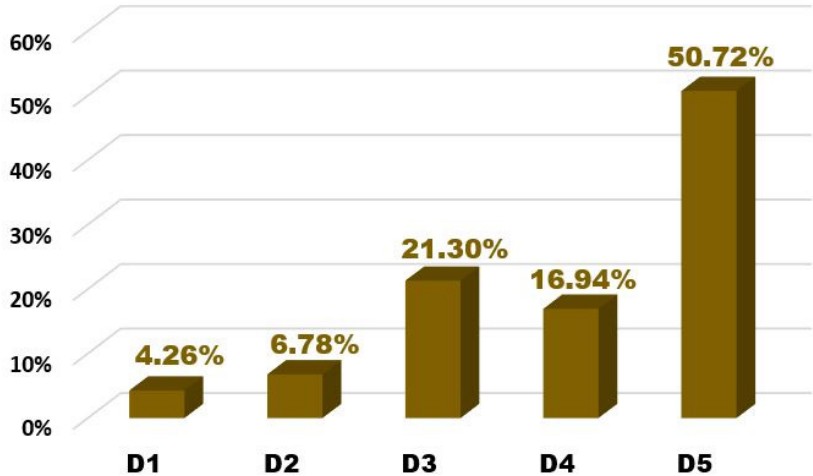

**Figure 4.** Percentage of presence by type of defect.

It is apparent that, as indicated by previous research [8–10], problems related to the undue presence of water in the envelope are the most common issues in auxiliary elements of facades (>70% defects), which is particularly significant in light of the high impact that this type of issue usually entails.

*3.3. Results by Type of Cause*

Among the 15 types of causes in this study, four stand out as a result of their frequency which, as shown in Figure 5, is greater than 10%. They are, in decreasing order of frequency: 'lack/anomaly of drip groove, gutter, and/or roof downpipes' (C01 = 15.68%), 'lack/insufficiency of sealant' (C02 = 13.85%), 'incorrect anchoring material' (C12 = 13.26%) and 'lack of barrier against capillary humidity' (C06 = 12.88%).

Three interesting causes reside in a lower frequency range (between 8% and 10%): 'characteristics of the material used' (C13 = 9.39%), 'direct contact with the terrain' (C07 = 8.81%), and 'lack/insufficiency of slope' (C03 = 8.71%).

Like the most common types of defects, the most common types of causes are closely related to water: causes C01, C02, C03, C04, and C06 are all related to water, humidity, and infiltrations, and they add up to more than 50% of causes.

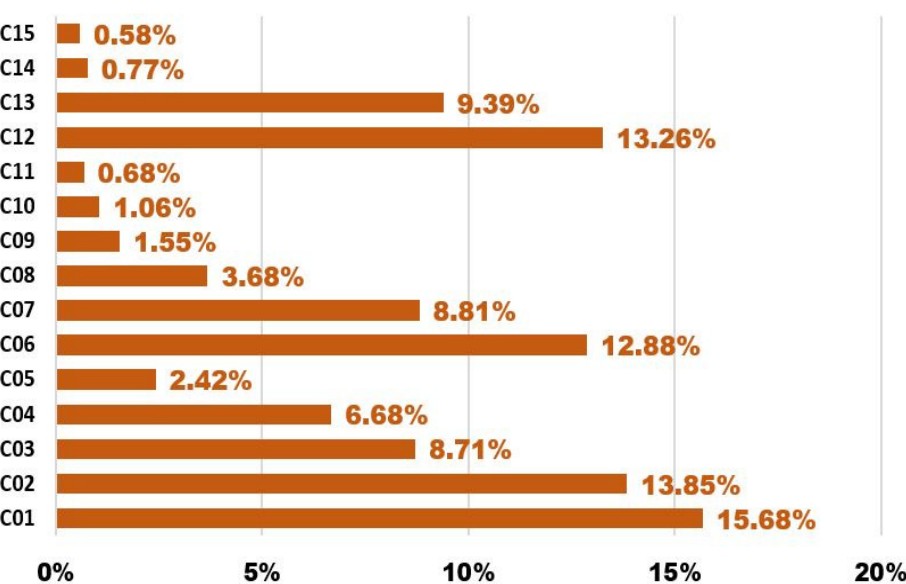

**Figure 5.** Percentage of presence by type of cause.

*3.4. Results by Type of Building*

The results by type of building (Figure 6) indicate that the overwhelming majority of cases (97.78%) occur in residential buildings (B1 + B2). Flats (B1) hold nearly 60% of cases, while houses (B2) account for nearly 40% of cases.

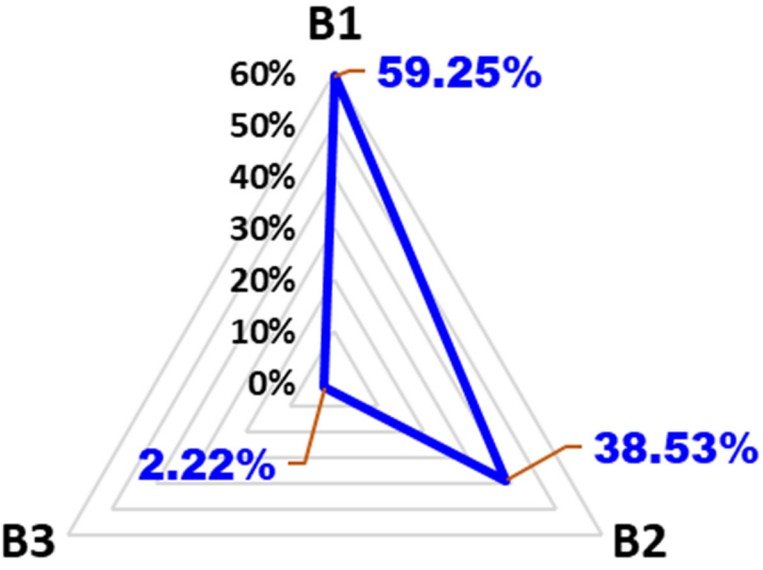

**Figure 6.** Percentage of presence by type of building.

*3.5. Results by Affinity Groups*

The results by types of affinity groups established in the methodology (D/AG and C/AG) are shown in Figure 7.

Figure 7 (left) shows the results related to the grouping of defects. It can be seen that the most common group is 'Humidities' (D/AG-H = 699 cases), whether as a consequence of 'rising damp' or 'infiltration humidity'. In second place is the 'Breaks' affinity group (D/AG-F = 290 cases).

Figure 7 (right) shows the results related to the grouping of causes. The most common group is 'Incorrect conditions of execution' (C/AG-I = 489 cases). The other two groups have very similar numbers of cases: C/AG-E = 296 cases and C/AG m = 248 cases.

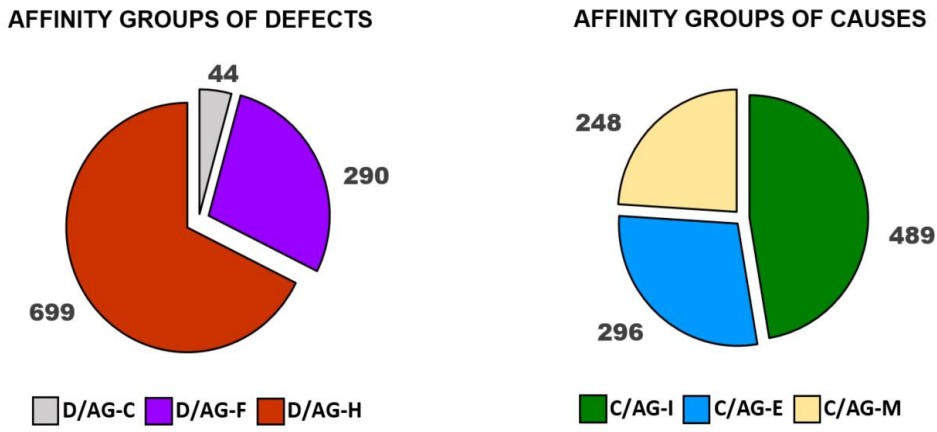

**Figure 7.** Number of cases by affinity groups.

When identifying the interrelations between the three affinity groups of defects and the three affinity groups of causes, we obtain the values shown in Table 3. It shows that the interrelation between C/AG-I (Incorrect conditions of execution) and D/AG-H (Humidities) is most pronounced, with 42.9%, followed by the interrelation between C/AG-E (Conditions of contact between elements) and D/AG-H (Humidities), with 23.8%.

**Table 3.** Matrix of percentage relations between the affinity groups of defects and causes.

| Causes \ Defects | D/AG-C | D/AG-F | D/AG-H |
|---|---|---|---|
| C/AG-I | 2.0% | 2.4% | 42.9% |
| C/AG-E | 0.5% | 4.4% | 23.8% |
| C/AG-M | 1.7% | 21.2% | 1.1% |

The percentage distribution within each type of element according to the affinity groups is shown in Figure 8. The most prevalent affinity group is D/AG-H (Humidities), which represents, with regard to defects, half the cases in E1 (cornices), two-thirds of cases E2 (external windowsills) and three-quarters of cases in E3 (exterior wainscots); nevertheless, D/AG-C (Cleaning) has very low values (between 3% and 8%). As for the affinity groups of causes, there is significant diversity in the percentages, though C/AG-I (Incorrect conditions of execution) represents the majority in E1 (cornices) and E2 (external windowsills), while C/AG-E (conditions of contact between elements) has the majority in E3 (exterior wainscots).

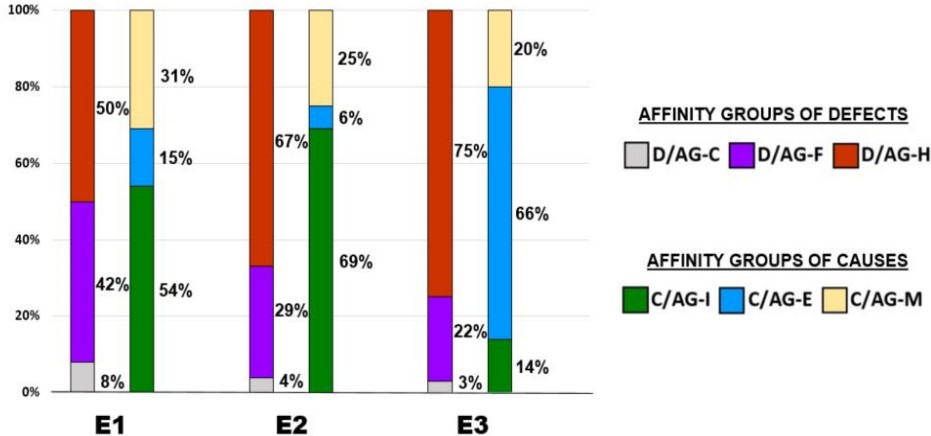

**Figure 8.** Percentage distribution of affinity groups by type of element.

### 3.6. Results by Type of Defect and Type of Building

With the objective of knowing the presence of each of the five types of defects across each of the three types of buildings, Table 4 was produced, showing the percentages expressed in relation to each of the defects analysed, as well to the total of the research. In this way, the highest percentage expressed in relation to the value of a defect is 'infiltration humidity—houses' (D5-B2 = 53.77%), while the highest percentage expressed in relation to the total of the research is 'infiltration humidity—flats' (D5-B1 = 28.85%).

**Table 4.** Percentages of presence according to types of defects and types of buildings (expressed in relation to the partial values of each type of defect and to the total set studied).

| Type of Defect | Types of Buildings | | | | | |
| --- | --- | --- | --- | --- | --- | --- |
| | B1 (%) | | B2 (%) | | B3 (%) | |
| | Over Type of Defect | Over Total Research | Over Type of Defect | Over Total Research | Over Type of Defect | Over Total Research |
| D1 | 5.07 | 3.00 | 3.27 | 1.26 | 0.00 | 0.00 |
| D2 | 8.01 | 4.74 | 4.77 | 1.84 | 8.70 | 0.19 |
| D3 | 26.31 | 15.59 | 13.32 | 5.13 | 26.09 | 0.58 |
| D4 | 11.93 | 7.07 | 24.87 | 9.58 | 13.04 | 0.29 |
| D5 | 48.68 | 28.85 | 53.77 | 20.72 | 52.17 | 1.16 |
| TOTAL | 100.00 | 59.25 | 100.00 | 38.53 | 100.00 | 2.22 |

The importance of defects related to humidities (D4 + D5) can also be noted in this table, in line with the rest of the data analysis carried out in this study. Nevertheless, it is worth indicating that D3 (detachment and fracture of pieces) holds a key position in B1 (flats) and B3 (other buildings).

### 3.7. Results by Element and Type of Defect or Cause

The distribution of cases according to the type of element was as follows: E1 = 118, E2 = 537, E3 = 378.

To know the type of defect (D) appearing in each element (E), Table 5 shows the correspondence between them (E–D). Based on this, we thus see that the highest number for 'element-defect' are: 'external windowsills—infiltration humidity' (E2–D5 = 359), 'external windowsills—detachment and fracture of pieces' (E2–D3 = 117), and 'exterior wainscots—infiltration humidity' (E3–D5 = 106), which when added together (582) represent over half of all cases. It should be mentioned that defect D4 (rising damp) only appears in E3 (exterior wainscots), while all other defects appear in all elements.

**Table 5.** Matrix of numerical relation between type of element and type of defect.

| E \ D | D1 | D2 | D3 | D4 | D5 |
| --- | --- | --- | --- | --- | --- |
| E1 | 9 | 19 | 31 | | 59 |
| E2 | 23 | 38 | 117 | | 359 |
| E3 | 12 | 13 | 72 | 175 | 106 |

The same procedure was carried out to know the type of cause (C) involved in each element (E). Table 6 shows the correspondence between them (E–C), indicating that the highest values occur in: 'exterior wainscots—lack of barrier against capillary humidity' (E3–C06 = 133), 'external windowsills—lack/anomaly of drip groove, gutter, and/or roof downpipes' (E2–C01 = 113), and 'external windowsills—lack/insufficiency of sealant' (E2–C02 = 106). It can also be seen that E1 has 7 different causes that lead to defects, E2 has

11 causes, and E3 has 13 different causes. The matrix shown in said table thus makes clear the important number of causes related to the appearance of humidities—the case that external windowsills and exterior wainscots are most affected, given their functions, which are related to water.

**Table 6.** Matrix of numerical relation between type of element and type of cause.

| E \ C | C01 | C02 | C03 | C04 | C05 | C06 | C07 | C08 | C09 | C10 | C11 | C12 | C13 | C14 | C15 |
|-------|-----|-----|-----|-----|-----|-----|-----|-----|-----|-----|-----|-----|-----|-----|-----|
| E1 | 46 | 5 | | 13 | | | | 16 | 2 | | | 14 | 22 | | |
| E2 | 113 | 106 | 90 | 56 | 6 | | | 21 | 4 | | 5 | 79 | 52 | 5 | |
| E3 | 3 | 32 | | | 19 | 133 | 91 | 1 | 10 | 11 | 2 | 44 | 23 | 3 | 6 |

*3.8. Relation between Type of Element, Type of Defect, and Type of Cause by Element and Type of Defect or Cause*

The total number of cases was broken down to show the distribution of damages-causes-elements and allow the characterisation of the data set. In this regard, Table 7 shows the causes behind each defect, as well as the element in which defects occur. This information is quite uncommon in construction research and can be of great interest for engineers and architects, as it allows them to pay greater attention to more problematic arrangements and take preventive measures accordingly.

At the same time, the breakdown in this table allows verifying the starting hypothesis presented by the authors in Table 2 (quadrant of construction interrelations). It is, for example, noted that there are no cases between D2 and C/AG-I, D4 and C/AG-I, or D4 and C/AG-M. It can also be noted that, in D/AG-C, D1 results from only 3 of the 15 possible types of causes; in D/AG-F, D2 results from 5 causes and D3 from 6 causes; in D/AG-H, D4 results from 4 different causes and D5 from 7 causes.

The starting hypothesis that D1-C01 (existence of spots and dirt—lack/anomaly of drip groove, gutter, and/or roof downpipes), D2-C13 (fissures originating in the construction process—characteristics of the material used), D3-C12 (detachment and fracture of pieces—incorrect anchoring material), and D4-C06 (rising damp—lack of barrier against capillary humidity) had a very recurrent interrelation is confirmed, given that they occur 4.8, 4.0, 5.5, and 7.6 times, respectively (in other words, they occur more than a third of the time).

In turn, interrelation D5-C02 (infiltration humidity—lack/insufficiency of sealant) was thought to be quite recurrent but turned out not to be, as it obtained a value of 2.7 times (less than a third of the time: it occurs in 143 out of the 524 total cases that occur in D5). This means that, with defect D5, there is no clearly predominant cause, as all causes have low recurrence (shown in Table 2 as 'O'). This is why D5-C02 has been listed in said table as 'p•/fO').

In turn, interrelation D2-C08 (fissures originating in the construction process—Lack of construction joints or expansion joints) was initially held as not recurring very much, as it was thought that the main interrelation with D2 would be D2-C13 (fissures originating in the construction process—characteristics of the material used), an aspect that was confirmed. We concluded that the interrelations with defect D2 as recurrent were: D2-C08 (3.3 times) and D2-C13 (4.0 times), for which reason they are indicated in Table 2 as 'pOf•' and '•', respectively. A similar situation took place with defect D1, in which it was thought that the most recurring interrelation would be D1-C01 (existence of spots and dirt—lack/anomaly of drip groove, gutter, and/or roof downpipes), something that was confirmed since a value of 4.8 times was obtained, but interrelation D1-C13 (existence of spots and dirt—characteristics of the material used) also emerged as recurrent, with a value of 4.1 times.

Once Table 7 was prepared and analysed, it was intended to verify the interrelation between elements, defects, and causes in different order. Figure 9 was thus prepared, showing that relation in a sequential hierarchy by: cause affinity group, type of cause, type of defect, and type of element.

**Table 7.** Matrix of numerical relation between the between type of element, type of defect, and type of cause.

| Table | Type of Cause | Type of Element | | | Subtotal | Total |
|---|---|---|---|---|---|---|
| | | **E1** | **E2** | **E3** | | |
| D1 | C01 | 5 | 16 | | 21 | 44 |
| | C09 | 2 | 1 | 2 | 5 | |
| | C13 | 2 | 6 | 10 | 18 | |
| D2 | C08 | 9 | 14 | | 23 | 70 |
| | C12 | 1 | 5 | 1 | 7 | |
| | C13 | 9 | 14 | 5 | 28 | |
| | C14 | | 5 | 3 | 8 | |
| | C15 | | | 4 | 4 | |
| D3 | C05 | | 6 | 19 | 25 | 220 |
| | C08 | 7 | 7 | 1 | 15 | |
| | C11 | | 5 | 2 | 7 | |
| | C12 | 13 | 67 | 40 | 120 | |
| | C13 | 11 | 32 | 8 | 51 | |
| | C15 | | | 2 | 2 | |
| D4 | C06 | | | 133 | 133 | 175 |
| | C07 | | | 23 | 23 | |
| | C09 | | | 8 | 8 | |
| | C10 | | | 11 | 11 | |
| D5 | C01 | 41 | 97 | 3 | 141 | 524 |
| | C02 | 5 | 106 | 32 | 143 | |
| | C03 | | 90 | | 90 | |
| | C04 | 13 | 56 | | 69 | |
| | C07 | | | 68 | 68 | |
| | C09 | | 3 | | 3 | |
| | C12 | | 7 | 3 | 10 | |
| | Sum | 118 | 537 | 378 | → | 1033 |

It can be seen that there are eight causes that lead to just one type of defect, four causes that lead to two types of defects, and three causes that lead to three types of defects. The lowest value for cause-defect-element is held by C15-D3 (two cases), followed by C09-D5 (three cases). The more numerous interrelations are C02-D5 (143 cases) and C01-D5 (141 cases). It can also be noted that the type of defect that occurs most often is D5 (seven times). This chart can be of great interest to professors of architecture and engineering, as it provides a causal summary of defects in external windowsills, exterior wainscots, and cornices.

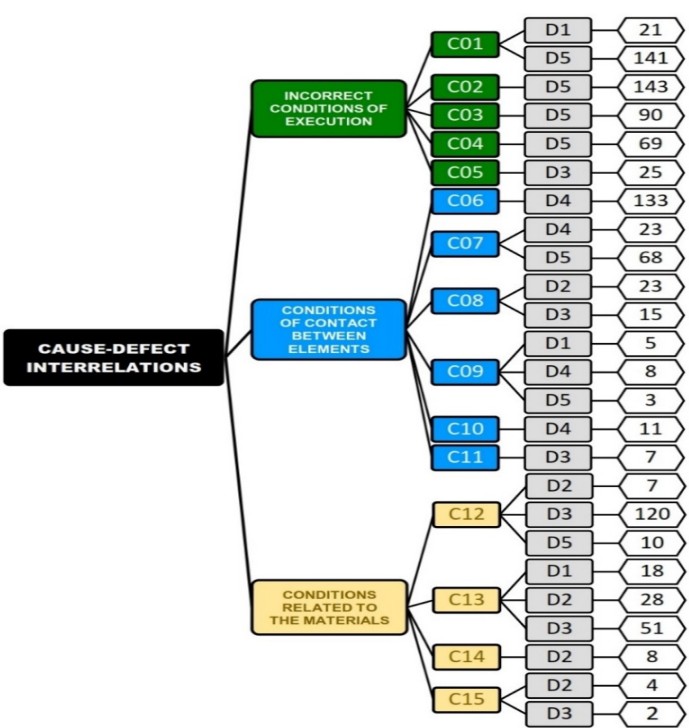

**Figure 9.** Descending tree chart of interrelations between types of causes and types of defects.

*3.9. Results by Type of Climate and Latitude*

The percentage distribution of cases, from the total set of defects, by type of climate and latitude, is as follows (see Figure 10). It can be seen that there is a higher concentration of defects in the Mediterranean climate and in the North latitude [51].

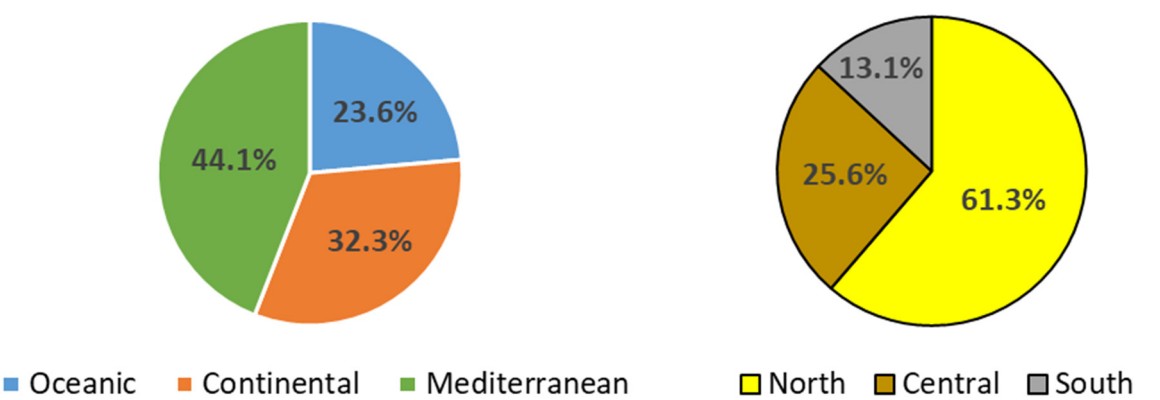

**Figure 10.** Distribution of cases by type of climate (**left**) and by latitude within Spain (**right**).

## 4. Discussion

*4.1. General Considerations*

1-As commented in the introduction, part of the defects in construction originates from inadequate or non-existent maintenance, which is quite a generalised problem in the Spanish building stock. For this reason, bringing maintenance activities up-to-date has gained increased importance over time [52]. Nevertheless, not all maintenance interventions fulfil their objectives well, which is why some research has even researched damages in facades [53] resulting from flawed conservation and renovation works in buildings [54].

2-The present research is aligned with the proposal put forth by previous studies [30] of creating a construction database that could be used to better identify construction defects.

This database could be maintained by a governmental agency of a private organisation with access by design engineers and site managers.

3-No other research was found in international scientific journals on the quantification of defects of auxiliary elements of facades whose data source consisted of lawsuits. It is thus considered that this is a novel contribution to the knowledge of the field of forensic engineering. The authors thus believe that the information and broken down indicators provided in this study facilitate a reduction in problems in construction designs and works, as well as a reduction in costs of non-quality.

4-In this context of complaints, it would be desirable to reduce litigation in cases that might have otherwise been resolved in a less traumatic manner. In Spain, where mediation in civil and commercial matters is voluntary, it would be a good idea to seek measures to incentivise and raise awareness about the use of mediation as a form of conflict resolution [55].

*4.2. Specific Considerations*

Although general results have been included by type of climate and latitude, the research team that conducted this paper continues with new lines of action, including the detailed breakdown of the possible interrelations and considerations between climate and defects. It should also be noted in this regard that no connection is observed between the defects of these three types of auxiliary elements and other parameters that could influence them if they were inside the dwellings, such as air conditioning or production of water vapor. This is because cornices, external windowsills, and exterior wainscots are elements that are completely on the outside of buildings' facades.

Based on the experience of previous works produced by the authors of this paper, it is considered that one of the best lessons learned pertains to the value of supervising the technical specifications included in the construction projects [56]. Being able to perform an in-depth assessment of the different components and areas of the facade drastically reduces the number of problems that may occur later in the service life of these facades. These conclusions were addressed in a publication on this aspect [57], in which 29 quality items (grouped by levels of severity) contained in the construction projects were verified.

**5. Conclusions**

The number of cases of defects analysed (1033) was significantly greater than what is commonly found in architectural engineering papers. All of the cases pertain to auxiliary elements of facades: external windowsills, cornices, and exterior wainscots. In these cases, five different types of defects and fifteen different types of causes were found and in turn grouped into three affinity groups, respectively. The respective type of building was also determined, and it was concluded that flats are the most numerous (B1 = 59.25%).

The element with the greatest presence of issues is 'external windowsills' (E2 = 51.98%). The most frequent type of defect is 'infiltration humidity' (D5 = 50.72%), and the most frequent type of cause is 'lack/anomaly of drip groove, gutter, and/or roof downpipes' (C01 = 15.68%). The most numerous defects affinity group is 'Humidities' (D/AG-H = 290 cases), and the most numerous causes affinity group is 'Incorrect conditions of execution' (C/AG-I = 489 cases).

The defect that occurs the most in a single element is D5 (infiltration humidity) in external windowsills (359 cases), while the cause that occurs the most is C06 (lack of barrier against capillary humidity) in exterior wainscots (133 cases). Likewise, the interrelation with the greatest number of cases between defect-cause is 'infiltration humidity—lack/insufficiency of sealant' (D5-C02 = 143 cases). Lastly, the confluence between defects affinity groups and causes affinity groups shows that the highest percentage of cases obtained is 'Humidities' with 'Incorrect conditions of execution' (D/AG-H and C/AG-I = 42.9%).

The results obtained in this study can be of interest and serve as a comparison for researchers of different countries that wish to know the outcomes of similar lawsuits in

their respective regions. It should be noted that the obtained values are 100% representative, as they correspond to the totality of cases in the period 2008–2017. All this can also help practically reduce the impact of works lacking in quality [58] in facades, which are one of the most exposed parts of buildings. Equally, the indicators included in this study will provide the various agents participating in building construction with very useful data to reduce defects [59]. Their knowledge will likewise lead to an increase in the sustainability of the construction process, given that there will be less litigation during the stage of use of buildings [60].

**Author Contributions:** Conceptualisation, M.J.C.-A.; methodology, M.J.C.-A.; validation, C.E.R.-J. and M.T.P.-A.; formal analysis, C.E.R.-J. and E.F.-T.; investigation, M.J.C.-A.; data curation, M.J.C.-A.; writing—original draft preparation, M.J.C.-A. and C.E.R.-J.; writing—review and editing, E.F.-T. and M.T.P.-A.; visualisation, M.T.P.-A. and E.F.-T.; supervision, M.J.C.-A. and C.E.R.-J. All authors have read and agreed to the published version of the manuscript.

**Funding:** This research received no external funding.

**Institutional Review Board Statement:** Not applicable.

**Informed Consent Statement:** Not applicable.

**Data Availability Statement:** Not applicable.

**Acknowledgments:** The current work was carried out within the Action Plan of the MUSAAT Foundation, in accordance with the 'Research project on building anomalies in Spain' [61].

**Conflicts of Interest:** The authors declare no conflict of interest.

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
