# Peer review of "Taxonomy of Defects in Auxiliary Elements of Facades and Its Relation with Lawsuits Filed by Property Owners"

_buildings, doi:10.3390/buildings12040401_

Round 1
Reviewer 1 Report
this is a very interesting paper to discuss building external windows and related lawsuit issues. As this paper is related to legal issues, and not a 100% scientific research paper, it would be good for the author to introduce the legal system and requirement related on buildings in the specific countries. Legal system and requirements, different from building science, vary from one country to another. A background of building related law requirements would be very helpful.
The data presented are interesting on defects and causes. It would be good for the paper to explain which types of data are related with weatherization and defects on window components, and which data/cases are related with the defects of window products or during installation of windows.
It would be interesting if the paper can present why the occupants want to suit the case, because of safety concerns, rain leakage, wind leakage or any other issues. It would be interesting to see what actually drive the occupants to notice the issues.
Also, the window types are also important, it would be interesting to see what cases are related with operable windows and if any defected are caused by window opening/close activities.
In the conclusion, it would be good the paper can summarize the experience learned from defects and how to avoid issues through O&M, installation, product defects and weatherization.
Author Response
Dear Reviewer,
We attach a file with the response to the review

Reviewer 2 Report
GENERAL COMMENTS
The paper deals with assessing the functional role of auxiliary elements of facades (focusing on exterior wainscots, cornices, and external windowsills) via the analysis of 1033 lawsuits filed in Spain for 10 years.
The study is very interesting and highly novel as it highlights the importance of these elements on the maintenance and performance of the building envelopes. The study is important for maintenance actions and planning new constructions.
The methodology used is logical and accurate, as well as the presentation of results. Only the Discussion section could have been better developed, as detailed below.
The paper is very well structured and written - it was a pleasure to read it.
Minor corrections and suggestions are listed below.
CORRECTIONS AND SUGGESTIONS
- 7, line 210: Please, substitute E1 by E2
- 14 – Figure 9: I think this figure could be better placed at the beginning of the Results section, as it illustrates the type of elements and common defects that are dealt with throughout the paper.
Discussion
This section is divided into 2 subsections: 4.1. General Considerations, and 4.2. Enneagram, architecture, and the eventual minimisation of defects. Though the contents of the second subsection are interesting, I don’t think that the topic merits a subsection in this paper as it is not directly discussing the results presented.
The authors have given sound comments along with the presentation of the results. I think that those comments can be used as the ground for a better overall discussion in this section.
Author Response
Dear reviewer,
We attach a file with the response to the review

Round 2
Reviewer 1 Report
The author has addressed the reviewer's comments.